behaviour/environmental science

daffodil cichlid, fish behaviour, man-made noise, noise pollution, playback experiment, social behaviour

**Author for correspondence:**
Andrew N. Radford
e-mail: andy.radford@bristol.ac.uk

# Impacts of additional noise on the social interactions of a cooperatively breeding fish

Ines Braga Goncalves[†], Emily Richmond[†], Harry R. Harding and Andrew N. Radford

School of Biological Sciences, University of Bristol, 24 Tyndall Avenue, Bristol BS8 1TQ, UK

HRH, 0000-0003-2511-3653; ANR, 0000-0001-5470-3463

Anthropogenic noise is a global pollutant known to affect the behaviour of individual animals in all taxa studied. However, there has been relatively little experimental testing of the effects of additional noise on social interactions between conspecifics, despite these forming a crucial aspect of daily life for most species. Here, we use established paradigms to investigate how white-noise playback affects both group defensive actions against an intruder and associated within-group behaviours in a model fish species, the cooperatively breeding cichlid *Neolamprologus pulcher*. Additional noise did not alter defensive behaviour, but did result in changes to within-group behaviour. Both dominant and subordinate females, but not the dominant male, exhibited less affiliation and showed a tendency to produce more submissive displays to groupmates when there was additional noise compared with control conditions. Thus, our experimental results indicate the potential for anthropogenic noise to affect social interactions between conspecifics and emphasize the possibility of intraspecific variation in the impacts of this global pollutant.

## 1. Introduction

Noise pollution, arising from human activities such as urbanization, resource extraction, infrastructure development and transportation, is prevalent in terrestrial and aquatic ecosystems across the globe. Additional noise can mask acoustic signals and cues, distract attention and cause a stress response, and thus has the potential to affect behaviour [1–3]. Indeed, in the last 15 years, there has been a rapidly expanding literature demonstrating noise impacts on, for instance, signalling, movement, foraging, vigilance, anti-predator responses and parental care, in a wide range of taxa [4–6]. However, these studies have tended to focus on the behaviour of individual organisms. Despite social interactions between conspecifics being crucial for many aspects of animal life, there has

[†]These authors contributed equally to the work

been relatively little experimental consideration of how these are affected by additional noise (for exceptions, see [7–10]).

In social species from ants to primates, conspecific outsiders threaten the resources (e.g. the territory, food, breeding positions and mating opportunities) of groups and their members [11–13]. Such outgroup conflict generates two important types of social interaction. First, there are interactions between a group and the outsider(s), which can range from information exchange through signalling contests to physical fights [14–16]. Second, outgroup conflict can influence within-group behaviour [17]. Recent studies have shown that encounters with outsiders or cues of their presence can lead to changes in affiliation and aggression among groupmates, either during or after the encounter [18–21]. Despite the ever-expanding literature on the impacts of anthropogenic noise [4–6], there has been no investigation of how acoustic disturbance may affect both group defence against conspecific intruders and the associated within-group interactions.

Here, we use a tank-based experiment on the cooperatively breeding cichlid *Neolamprologus pulcher*—a model fish species that exhibits natural behaviours in captivity [22]—to investigate how additional noise affects the defensive actions displayed towards an intruding outsider and on the concurrent social interactions among groupmates. *Neolamprologus pulcher* has occasionally been referred to as *N. brichardi* in the literature [23]; however, they are currently considered closely related but separate species [24]. *Neolamprologus pulcher* live in groups comprising a dominant breeding pair and 0–20 subordinate helpers who aid in the protection and care of clutches and young fry [25,26]. The species is territorial, with defence regularly displayed against outsiders challenging for dominance and breeding positions; all group members can participate in defensive actions [27,28]. Within-group acts of affiliation, aggression and submission are commonplace [27,29] and are more common during and following conflict with conspecific intruders than at other times [20,30]. *Neolamprologus pulcher* has been demonstrated to detect sounds between 100 and 2000 Hz [31], but is also capable of producing high-frequency sounds (two-pulsed calls with a peak pulse frequency of $12 \pm 8$ kHz) for communication [32]; the species is, therefore, likely to have a hearing capacity to match. Moreover, previous work has shown that *N. pulcher* behaviour in contexts unrelated to outgroup conflict can be affected by playback of additional noise [7]. We predicted that additional noise would either act as a distraction, resulting in a reduction in social behaviour, or act as a stressor and so potentially result in increased aggression towards either the intruder and/or groupmates.

# 2. Material and methods

## 2.1. General set-up

We conducted our study using a captive population of *N. pulcher* housed at the University of Bristol. Study groups each comprised three individuals: a dominant female (DF), a dominant male (DM) and a subordinate female (SF). As size is a good indicator of dominance in this species [22,33], SFs were a minimum of 9 mm shorter in standard length than the DF in their group. Where possible, the SF and DF were related (determined from previous group formations in earlier studies) to reduce aggression and to increase group stability [22]. Once formed, we closely monitored groups for a minimum of two weeks prior to the experiment to ensure a stable hierarchy had formed. This was evidenced by general group cohesion and a lack of sustained aggressive behaviour towards particular individuals [20,28]. We also checked daily for new clutches. Three groups produced a clutch prior to the start of their trials; we removed the clutches and waited at least 3 days before conducting the experiment on these groups. None of the groups produced clutches during their trial period.

We housed each fish group in an individual 70 l tank (width × length × height: $30 \times 61 \times 38$ cm), which formed its territory (as per [20,30]). Tanks were each equipped with 2–3 cm of sand (Sansibar river sand), a 75 W heater (Eheim), a filter (Eheim Ecco pro 130), a thermometer (Eheim), two plant-pot halves (each 10 cm wide) for shelters, a tube shelter and artificial plants. For all experimental trials, we removed the filter inlet and outlet, as well as the heater, plants and tube shelter, to allow easy observation during video analysis. The two plant-pot shelters were left in place to maintain the familiar breeding territory. We placed tanks on polystyrene sheets to reduce noise transfer and disturbance effects. Tanks were also visually isolated from one another by opaque ViPrint sheets (0.35 mm thickness) that surrounded all tank walls excluding the wall facing into the room. We kept water temperature at $27 \pm 1°C$ (mean ± s.e.) and set room lights on a 13L : 11D hour cycle (daylight from 7.00 to 20.00) to simulate the natural light cycle around Lake Tanganyika. We fed fish twice

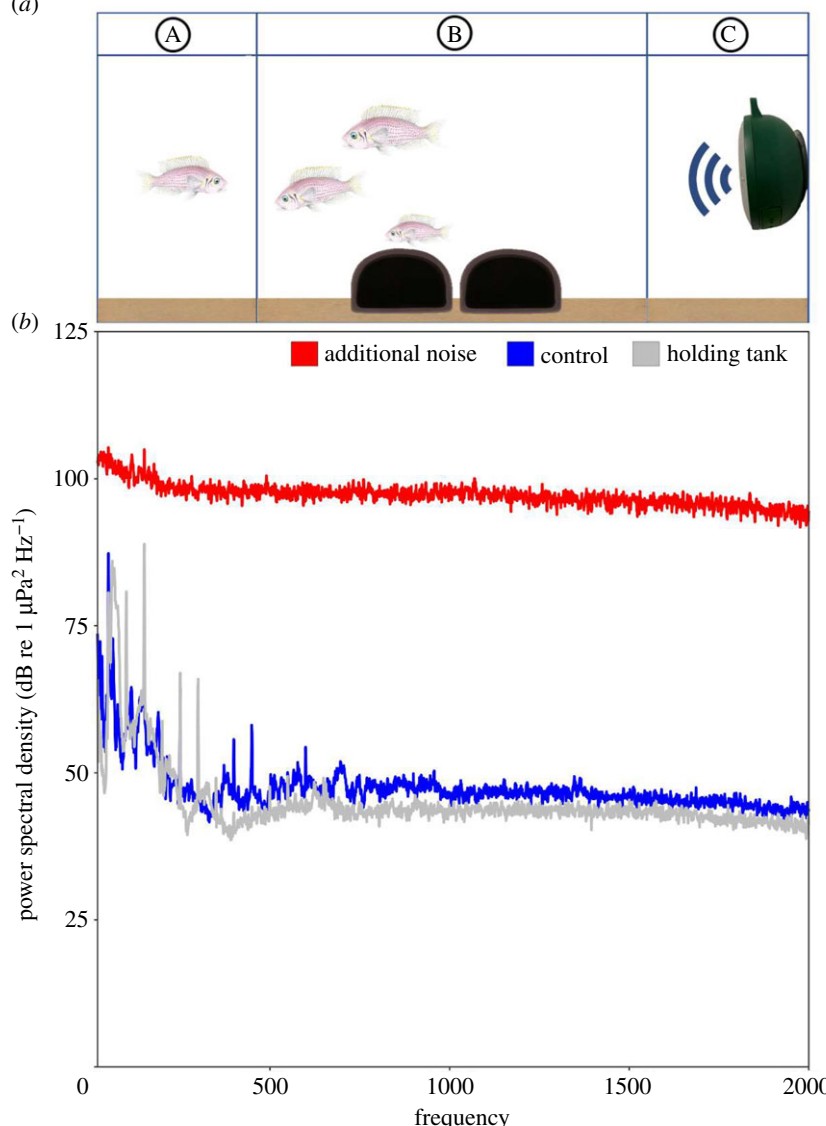

**Figure 1.** Schematic of experimental set-up (*a*) and representative power spectral densities of sound-pressure levels (*b*). In (*a*), the three compartments of resident group tanks are shown; the focal group were always in compartment B; the intruder and the loudspeaker could be in either compartment A or compartment C. Fish drawing by Martin Aveling. In (*b*), sound-pressure levels are shown for additional-noise playback (red), silent control (blue) and holding-tank conditions (grey). Recordings were taken in the centre of the tank, 10 cm from the bottom, and analysed using Matlab acoustics-analysis package paPAM [34], FFT length = 48 000, Hamming evaluation window, 50% overlap, 20 s recordings (equivalent to one period of additional-noise playback).

daily: a combination of frozen brine shrimp, water fleas, prawns, mosquito larvae, mysid shrimp, bloodworms, cichlid diet, spirulina, copepods, krill and sludge worms in the mornings on Monday to Friday; and dry fish flakes in the evenings and weekends.

We could separate each tank into three sections by sliding opaque white ViPrint partitions (75 mm) through single-channel PVC tracks glued to the inside of the long walls, 8 cm from the tank edge at either end. This created two side compartments measuring 30 × 8 cm (width × length) each and a larger central compartment (30 × 45 cm). During experimental trials, one side compartment housed the intruding female (with sufficient space for her to move freely; [20]) and the other contained the underwater loudspeaker; the resident group was in the central compartment (figure 1*a*).

## 2.2. Experimental details

We gave 16 groups two treatments each in a repeated-measures design: during the simulated territorial intrusion of a rival female, there was playback of either white noise (additional noise) or silence (as a

control). The intruding female was size-matched (within 2 mm) to the resident DF [20]; the same female intruder was used for both treatments to the same focal group, but different intruders were used for each focal group. We performed trials to the same group at least 24 h apart, and counterbalanced treatment order between groups. No group used in this experiment had previously been exposed to additional-noise playback.

We generated additional-noise playback tracks (one for each focal group) in Audacity (v. 2.3.0), and applied a high-pass filter of 100 Hz so that they played within the frequency range of the underwater loudspeaker (as per other playback studies; see [7]). Playback tracks were each 12 min long, comprising repeating blocks of 5 s silence, 20 s white noise, 5 s silence; this resulted in 40s of white noise and 20 s of silence per minute. Each white-noise period had a 10s fade-in and 10 s fade-out effect applied to reflect the temporal fluctuation in amplitude of, for example, a passing motorboat. Previous studies have also used artificial noise (e.g. white noise) to assess animal responses to acoustic disturbance; doing so allows greater standardization and avoids pseudoreplication of the playback of one or a few exemplars of anthropogenic-noise sources [35,36]. During control trials, we played back a silent track to control for any magnetic field generated by an active loudspeaker. The underwater loudspeaker (Aqua-30, DNH, Norway; effective frequency range 0.08–20 kHz) was connected to an amplifier (M033N, 18 W, frequency response 0.04–20 kHz; Kemo Electronic GmbH) and a battery (12 V, 12 Ah, sealed lead-acid), and controlled through a laptop (MacBook Air 2015).

We determined representative power spectral densities from sound-pressure recordings made using a calibrated omnidirectional hydrophone (HiTech HTI-96-MIN with inbuilt preamplifier, manufacturer-calibrated sensitivity −164.3 dB re 1 V/µPa; frequency range 0.02–30 kHz; High Tech Inc., Gulfport, MS) and digital recorders (PCM-M10, 48 kHz sampling rate, Sony Corporation, Tokyo, Japan; H1n, 96 kHz sampling rate, Zoom Corporation, Tokyo, Japan). Fish are also sensitive to the particle-motion component of sound fields [34], but we were unable to measure this domain due to the unavailability of relevant equipment that can make accurate measurements in small tanks. All sound analysis was conducted in Matlab (r2013a) using extension software PaPAM (v. 0.872; [36]). Initially, to determine relative sound levels in different tanks conditions—for the acoustic treatments (additional-noise playback and the silent control treatment) and the holding tank with the filter and heater turned on— we positioned the hydrophone in the centre of the tank, approximately 10 cm from the bottom and 20 cm from the loudspeaker. In the frequency band 20–2000 Hz (figure 1b), fish were exposed to a root-mean-square (RMS) sound-pressure level (SPL) in dB re 1 µPa of 96.9 (holding tank), 90.5 (silent control) and 132.2 (additional noise), as determined from 20 s recordings (equivalent to one period of additional noise); there was little variation between the 16 additional-noise playback tracks used in the trials (range 131.4–132.8 dB re 1 µPa). To account for the possible detection of high-frequency sounds [32], the RMS SPL over the frequency band 4–20 kHz was also calculated in dB re 1 µPa: 82.4 (holding tank), 81.6 (silent control) and 148.1 (additional noise). Having found (see Results) that additional noise affected some within-group interactions (which can occur anywhere in the focal group compartment), but not defensive actions against the intruder (which occur next to the intruder compartment, furthest from the loudspeaker), we made sound-level measurements in the corners next to the intruder compartment and those next to the loudspeaker compartment (10 cm from the bottom, to match those made in the centre). While RMS SPL levels in the frequency range 0.02–2 kHz were lower closest to the intruder compartment (127.7–129.5 dB re 1 µPa) than in the centre or nearest the loudspeaker (130.0–132.9 dB re 1 µPa), they were still considerably higher than in the control treatment and probably detectable by the fish.

Experimental trials followed the general intrusion protocol of [20]. We conducted all simulated intrusions between 9.00 and 12.30, to minimize natural daily variations in behaviour, hormone levels and hunger [37]. Prior to the start of a trial, we slid a transparent partition next to the opaque partition on the tank side where the intruding female would be introduced. The transparent partition meant that the resident group and the intruder could visually interact, while avoiding any opportunity to cause harm through direct aggression. We then netted the intruder from her home tank, placed her directly into the relevant side compartment of the focal tank and left her to settle out of sight of the resident group for at least 5 min. We started the playback 1 min before the end of the settling period. One minute later, we removed the opaque partition to allow visual interaction between the intruder and the resident group, and the 10 min data-collection period began. At the end of the intrusion, we replaced the opaque partition and netted the intruding female for immediate return to her home tank. Females from a tested focal group were not used as an intruder for a minimum of one week. To limit the potential effects of disturbance by acoustic transfer between neighbouring tanks (SPL of additional-noise playback recorded in an adjacent tank to the one in

which the loudspeaker was active: 20–2000 Hz, SPL = 98.9–103.0 dB re 1 µPa), the groups in adjacent tanks to a focal group were not tested for a minimum of a week after trials to the latter had finished. We video-recorded (Sony Handycam HDR-XR520) all trials, from the point that the intruder and resident group could interact visually, for later collection of behavioural data.

## 2.3. Data analysis

We saved all trial videos with coded file names and watched them without sound to ensure behavioural scoring was completed blind to acoustic treatment. ER used BORIS (v. 7.4.7 windows XP) to score behaviours based on the ethogram from [20], which was originally formed using previously established protocols for the study species [38–40]. For each 10 min intrusion period, we scored defensive behaviours (ramming, biting, frontal displays and aggressive postures directed towards the intruder or connecting with the transparent partition); we scored the total amount received by the intruder and that exhibited by each category of individual (DM, DF, SF). We also scored all within-group social interactions during the intrusion period as aggressive (ramming, biting, frontal displays, chasing and aggressive posturing), affiliative (soft-touch, parallel swimming, following and joining) and submissive ('hook and J' displays, quivering and head-up postures).

For all datasets, we assessed residual distribution visually with Q–Q plots and statistically with Shapiro–Wilk tests. When residuals were normally distributed, we used parametric tests; otherwise, we used non-parametric equivalents. All tests were conducted using RStudio v. 1.3.1093. During analysis, one group was identified as having two breeding females, and so was not subsequently considered, resulting in data from 15 groups for all analyses. Treatment order did not significantly affect any of the four measured behaviours (paired $t$-tests, defence: $t = 0.84$, d.f. = 14, $p = 0.414$; submission: $t = 0.66$, d.f. = 14, $p = 0.523$; Wilcoxon tests, aggression: $V = 64$, $n = 15$, $p = 0.887$; affiliation: $V = 62$, $n = 15$, $p = 0.262$), so we used paired $t$-tests or Wilcoxon tests to analyse differences between acoustic treatments (additional noise and control). We first examined the effect of additional noise on the total defensive effort against the intruder; we then determined whether the non-significant treatment difference (see Results) was consistent across all three individual categories or if there were counterbalancing effects between group members. Second, we investigated the effect of additional noise on the overall amount of within-group aggression, affiliation and submission exhibited; we used the sequential Bonferroni correction as there was a separate test for each behaviour. For those behaviours found to be significantly different between treatments (affiliation and submission; see Results), we determined which group members were driving the differences. Summary treatment values for all behaviours are provided in table 1; see [41] for datasets.

# 3. Results

Groups directed an average of $51.4 \pm 28.0$ (mean ± s.d.; range: 10–115) aggressive acts towards the intruder during a trial. However, there was no significant difference between the additional-noise and control treatments in the total amount of aggression displayed (paired $t$-test: $t = 1.04$, d.f. = 14, $p = 0.320$). All three categories of individual showed a similar lack of treatment difference in defensive behaviour (DF: $t = 0.34$, d.f. = 14, $p = 0.742$; DM: $t = 1.84$, d.f. = 14, $p = 0.087$; SF, Wilcoxon test: $V = 48.5$, $n = 15$, $p = 0.531$).

During a trial, there was an average of $9.8 \pm 14.0$ (range: 0–69) aggressive acts, $4.9 \pm 5.5$ (range 0–29) affiliative acts and $10.8 \pm 8.7$ (range: 1–40) submissive acts between groupmates. While acoustic treatment did not significantly influence the amount of within-group aggression (Wilcoxon test: $V = 35$, $n = 15$, $p = 0.163$), it did significantly affect both within-group affiliation ($V = 82$, $n = 15$, $p = 0.012$; adjusted $\alpha = 0.025$) and submission (paired $t$-test: $t = 2.95$, d.f. = 14, $p = 0.010$; adjusted $\alpha = 0.0167$). Compared with control conditions, additional noise resulted in less affiliation (figure 2a) and more submission (figure 3a).

The noise-induced changes in within-group behaviour were driven by effects on the female group members. The lower level of affiliation in the additional-noise treatment was the consequence of a significantly smaller amount by both DFs (Wilcoxon test: $V = 57$, $n = 15$, $p = 0.035$; figure 2b) and SFs ($V = 21$, $n = 15$, $p = 0.031$; figure 2b) when compared with the control treatment; there was no significant treatment difference in DM affiliative behaviour ($V = 52$, $n = 15$, $p = 0.282$). Both DFs (paired $t$-test: $t = 2.01$, d.f. = 14, $p = 0.064$; figure 3b) and SFs ($t = 2.06$, d.f. = 14, $p = 0.059$; figure 3b) exhibited more submissive behaviour when there was additional noise compared with control conditions, although neither result was statistically significant; DMs were hardly ever submissive towards DFs in either treatment.

**Table 1.** Summary statistics (mean ± s.d., range) for the number of each type of behaviour displayed by the group as a whole (group) and by each of the dominant female (DF), dominant male (DM) and subordinate female (SF). Presented are values for those behaviours analysed in the main text.

| behaviour | group or individual | control | noise |
|---|---|---|---|
| defence | group | 49.1 ± 30.6 | 53.7 ± 26.0 |
| | | (10–115) | (16–104) |
| | DF | 21.6 ± 25.4 | 20.5 ± 21.8 |
| | | (0–81) | (1–76) |
| | DM | 21.5 ± 16.8 | 27.3 ± 20.8 |
| | | (0–61) | (5–72) |
| | SF | 5.9 ± 5.9 | 5.9 ± 4.5 |
| | | (0–20) | (0–15) |
| affiliation | group | 6.6 ± 7.0 | 3.1 ± 2.6 |
| | | (1–29) | (0–8) |
| | DF | 2.5 ± 3.1 | 0.7 ± 1.1 |
| | | (0–11) | (0–3) |
| | DM | 2.9 ± 4.0 | 1.9 ± 2.3 |
| | | (0–13) | (0–8) |
| | SF | 1.2 ± 1.7 | 0.5 ± 1.1 |
| | | (0–5) | (0–4) |
| aggression | group | 8.2 ± 11.5 | 11.5 ± 16.5 |
| | | (0–46) | (0–69) |
| submission | group | 9.1 ± 8.1 | 12.3 ± 9.3 |
| | | (1–33) | (3–40) |
| | DF | 2.9 ± 3.3 | 3.9 ± 3.7 |
| | | (0–12) | (0–11) |
| | SF | 6.3 ± 8.7 | 8.3 ± 9.7 |
| | | (0–33) | (2–40) |

## 4. Discussion

Using a captive-based playback experiment on the cichlid fish *N. pulcher*, we found that additional noise did not have a significant impact on the amount of aggressive defence displayed towards a conspecific intruder, but that some associated within-group interactions were affected. There was no sound-treatment effect on within-group aggression, but additional noise resulted in significantly less within-group affiliation and more submission between groupmates. These noise-induced changes in behaviour were driven by the responses of the female group members. We thus provide experimental evidence for an effect of acoustic noise on social interactions between conspecifics, as well as intraspecific variation in noise responses.

The lack of a noise effect on cichlid defensive actions contrasts findings in various other fish and bird species, which have shown either reduced [42–45] or elevated [46] aggressive responses when there is additional noise. It is unlikely that our result is due to a lack of noise detection by the cichlids for several reasons: within-group behaviours were affected; an earlier study found behavioural impacts of additional noise in *N. pulcher* [7]; and there is evidence that this species produces sounds for communication [32] and sound levels near the intruder compartment were still probably audible to the focal fish. Instead, there are potentially important differences between our experiment and previous noise-related studies on defensive behaviour, as the latter focused on vocal indicators of territorial intrusions and considered solitary and pair-bonded species [42–45]. Vocalizations can be masked by additional noise [1], whereas we provided a visual stimulus. While there can be cross-modal effects of noise on the use of sensory information from other modalities [47], perhaps the physical presence of an intruder is a sufficiently

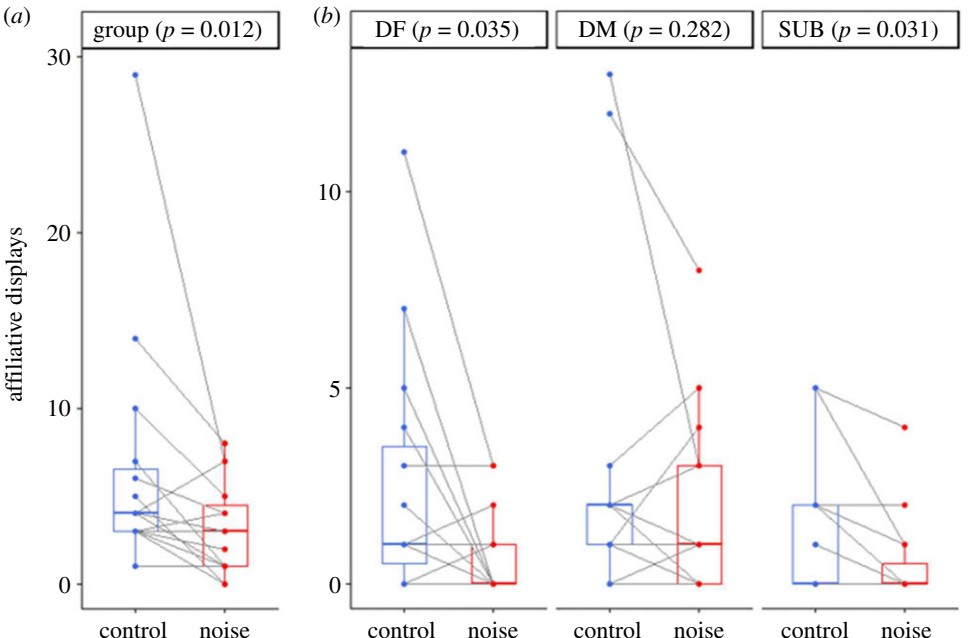

**Figure 2.** Within-group affiliative behaviour, during 10 min territorial intrusions of a rival dominant female, when there was playback of either white noise (noise) or silence (control). (a) Total number of affiliative displays by all group members; (b) affiliative displays by the dominant female (DF), dominant male (DM) and subordinate female (SF). Boxplots display medians, 25% and 75% quartiles and dots represent raw data. $N = 15$ individuals in all cases, although data values for some individuals are the same.

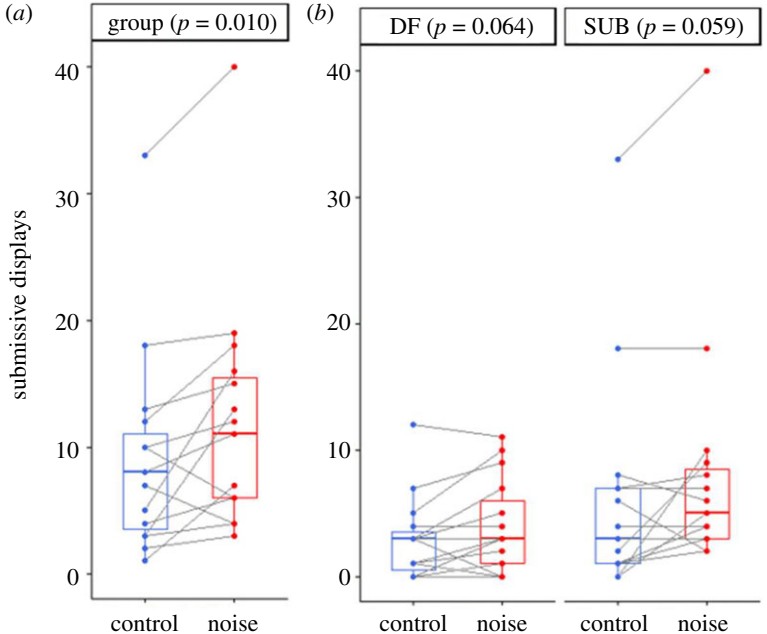

**Figure 3.** Within-group submissive behaviour, during 10 min territorial intrusions of a rival dominant female, when there was playback of either white noise (noise) or silence (control). (a) Total number of submissive displays by all group members; (b) submissive displays by the dominant female (DF) and subordinate female (SF); the dominant male hardly ever displayed submissively in either treatment. Boxplots display medians, 25% and 75% quartiles and dots represent raw data. $N = 15$ individuals in all cases, although data values for some individuals are the same.

strong stimulus to mean that noise disruption is less likely. Context is known to influence responsiveness to noise [48], including in the study species [7], and certain external stimuli may demand such a strong focus that there might be a lessening of, at least, the distracting aspect of noise. Moreover, it is possible that the defensive behaviour of group-living species is less affected by noise due to reinforcement between group members that sustains the overall level.

Our finding that additional noise had mixed impacts on within-group interactions is in line with an earlier study on the same species that considered different behavioural contexts [7]. We found overall increases in submission, driven by both the DF and SF group members; the previous study also found a noise-induced increase in subordinate submission [7]. Submission is energetically costly, increasing routine metabolic rate more than threefold in the study species [49], which could detrimentally affect the payoffs relating to group membership and dispersal decisions [50]. We also found a female-driven decrease in within-group affiliation; the previous study did not examine affiliation or intersexual differences [7]. A reduction in affiliation could disrupt relationships and increase conflict within groups [51,52]. The stronger noise effect on females compared with males might be because the intruder was a female; she was size-matched to the DF of the focal group, who might, therefore, have been under more threat than the DM. Subordinate females may have altered their behaviour in response to the observed changes in DF behaviour as well as to the sound treatment. For instance, DFs have been shown to direct significantly less affiliation toward SFs when threatened by more active large female intruders [20]. Equally, there might be some consistent intraspecific variation in noise impacts [48,53]. Further experimentation will be needed to tease apart these possibilities, to determine exactly which noise frequencies drive the effects and to assess whether the changes in behaviour are the response of fish to a possible competitor under noisy conditions or the response to noisy conditions *per se*.

In social animals, most daily activities involve interactions with conspecifics. Using a model captive system and sound playback, we have demonstrated the potential for anthropogenic noise to disrupt some of these social interactions. However, it is important that future work considers whether these effects remain with repeated or chronic noise exposure [54] and tests the influences on social interactions in natural conditions with real noise sources [53]. If social interactions are indeed affected by anthropogenic noise, then potential consequences for territoriality, conflict resolution and within-group dynamics can be added to the growing list of impacts of this global pollutant.

Ethics. All procedures were conducted with permission from the University of Bristol Ethical Committee (University Investigator Number: UB/16/049).

Data accessibility. All data are available from the Dryad Digital Repository: https://doi.org/10.5061/dryad.fxpnvx0pn [41].

Authors' contributions. I.B.G., E.R and A.N.R. conceived the project. I.B.G. and E.R. collected the behavioural data. H.R.H. prepared the sound files and made the acoustic recordings. I.B.G. and E.R. analysed the behavioural data; H.R.H. analysed the acoustical data. I.B.G., E.R. and A.N.R. interpreted study findings. A.N.R. drafted the paper. All authors contributed to editing drafts of the manuscript and approved the final draft. All authors agree to be held accountable for the work performed therein.

Competing interests. We declare we have no competing interests.

Funding. This work was supported by a European Research Council Consolidator Grant (project no. 682253) awarded to A.N.R.

Acknowledgements. We thank Amy Morris-Drake and Patrick Kennedy for the useful discussion about the experiment, Barbara Taborsky and Michael Taborsky for valuable information about the study species, and Martin Aveling for the beautiful fish artwork in figure 1.

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
