## [Peer Review File · Royal Society Open Science]

Review History

RSOS-202191.R0 (Original submission)

Review form: Reviewer 1

Is the manuscript scientifically sound in its present form?

No

Are the interpretations and conclusions justified by the results?

No

Is the language acceptable?

Yes

Do you have any ethical concerns with this paper?

No

Have you any concerns about statistical analyses in this paper?

No

Recommendation?

Reject

Comments to the Author(s)

While I appreciate the authors attempts to remedy some of my comments, I still have two concerns.

I do not think the request to sound map more of the tank was unreasonable and could easily be accomplished. If the fish never moved from the spot where the sound was recorded, than that would be fine however the authors will need to show a resident time map of the fish location. The high level of variability shown in the study could be partially explained by large/sharp sound gradients in close proximity to the location where the sound was recorded.

The use of broad band sound from 0.1 to 20 Khz still remains unjustified. While I may agree that some sound caused some behavioral differences, I think it would be reasonable to understand what frequencies may have influenced behavior. While I agree that ABRs have limited value, they are certainly useful to establish what frequencies that the fish could detect. The paper that reports this fish produces sound up to 12Khz plots the sound spectrum from 4 to 12kHz. And while one would think that sounds produce by animals are effective for intraspecific communication, this does not necessarily mean that the entire spectrum is detected. Additionally, these sounds are exceedingly short (10 ms) and their functional significance remains to be determined.

Without compelling physiological or behavioral evidence to clearly indicate that species can detect frequencies above 5 Khz, most of the sound presentation in the frequencies < 5 Khz was limited to 100 dB which is probably why the behavioral differences were not clear. Also, the sound is claimed to be representative of transient motorboat but there is no evidence that this white noise is consistent with the spectrum from a boat motor. The paper would be considerably strengthened by focusing on the frequencies that are within the sensitivity range of this and other species and examining the effects of different sound amplitudes on the behavior.

Review form: Reviewer 2**Is the manuscript scientifically sound in its present form?**

Yes

Are the interpretations and conclusions justified by the results?

Yes

Is the language acceptable?

Yes

Do you have any ethical concerns with this paper?

No

Have you any concerns about statistical analyses in this paper?

No

Recommendation?

Accept with minor revision (please list in comments)

Comments to the Author(s)

This is a nice short paper in which the authors experimentally manipulated the acoustic environment and showed that the addition of white noise did not change the aggressive responses of cichlid groups towards a possible intruder, but did reduce affiliation and increased submissive behaviors of females within the groups. I found the paper easy to read and interesting, and the topic timely.

I have only one comment – the experimental design makes it impossible to tease apart the response of the fish to a possible competitor under noisy conditions from the response to the noisy conditions per se. The noise playback always started just before the fish were exposed to the competitors, and theoretically the reduced affiliation and increased submissive behavior could be a reaction of the fish to the new noisy conditions, regardless of the intruding fish. This can be at least partly dealt with by looking at the behavior of the group members in the minute that passed between the starting of the playback and the exposure the fish to the intruder.

Decision letter (RSOS-202191.R0)

Dear Professor Radford

The Editors assigned to your paper RSOS-202191 "Impacts of additional noise on the social interactions of a cooperatively breeding fish" have made a decision based on their reading of the paper and any comments received from reviewers.

Regrettably, in view of the reports received, the manuscript has been rejected in its current form. However, a new manuscript may be submitted which takes into consideration these comments.

We invite you to respond to the comments supplied below and prepare a resubmission of your manuscript. Below the referees' and Editors' comments (where applicable) we provide additional requirements. We provide guidance below to help you prepare your revision.

Please note that resubmitting your manuscript does not guarantee eventual acceptance, and we do not generally allow multiple rounds of revision and resubmission, so we urge you to make every effort to fully address all of the comments at this stage. If deemed necessary by the Editors, your manuscript will be sent back to one or more of the original reviewers for assessment. If the original reviewers are not available, we may invite new reviewers.

Please resubmit your revised manuscript and required files (see below) no later than 30-Aug-2021. Note: the ScholarOne system will 'lock' if resubmission is attempted on or after this deadline. If you do not think you will be able to meet this deadline, please contact the editorial office immediately.

Please note article processing charges apply to papers accepted for publication in Royal Society Open Science (<https://royalsocietypublishing.org/rsos/charges>). Charges will also apply to papers transferred to the journal from other Royal Society Publishing journals, as well as papers submitted as part of our collaboration with the Royal Society of Chemistry (<https://royalsocietypublishing.org/rsos/chemistry>). Fee waivers are available but must be requested when you submit your manuscript (<https://royalsocietypublishing.org/rsos/waivers>).

Thank you for submitting your manuscript to Royal Society Open Science and we look forward to receiving your resubmission. If you have any questions at all, please do not hesitate to get in touch.

on behalf of Dr Isaac Ligocki (Associate Editor) and Kevin Padian (Subject Editor)
openscience@royalsociety.org

Associate Editor Comments to Author (Dr Isaac Ligocki):

Comments to the Author:

The authors exposed groups of the cooperatively breeding fish *Neolamprologus pulcher* to white noise, and found that white noise did not influence aggressive behavior towards a conspecific intruder, but that white noise did influence within group interactions (affiliative and submissive behaviors). The authors have made substantial changes to the manuscript in response to the comments of previous reviewers. Reviewers highlighted several remaining issues with the updated manuscript. One primary area of concern expressed by R1 relates to the broad range of sound utilized in the study.

Reviewer comments to Author:

Reviewer: 1

Comments to the Author(s)

While I appreciate the authors attempts to remedy some of my comments, I still have two concerns.

I do not think the request to sound map more of the tank was unreasonable and could easily be accomplished. If the fish never moved from the spot where the sound was recorded, than that would be fine however the authors will need to show a resident time map of the fish location.

The high level of variability shown in the study could be partially explained by large/sharp sound gradients in close proximity to the location where the sound was recorded.

The use of broad band sound from 0.1 to 20 Khz still remains unjustified. While I may agree that some sound caused some behavioral differences, I think it would be reasonable to understand what frequencies may have influenced behavior. While I agree that ABRs have limited value, they are certainly useful to establish what frequencies that the fish could detect. The paper that reports this fish produces sound up to 12Khz plots the sound spectrum from 4 to 12kHz. And while one would think that sounds produce by animals are effective for intraspecific communication, this does not necessarily mean that the entire spectrum is detected.

Additionally, these sounds are exceedingly short (10 ms) and their functional significance remains to be determined.

Without compelling physiological or behavioral evidence to clearly indicate that species can detect frequencies above 5 Khz, most of the sound presentation in the frequencies < 5 Khz was limited to 100 dB which is probably why the behavioral differences were not clear. Also, the sound is claimed to be representative of transient motorboat but there is no evidence that this white noise is consistent with the spectrum from a boat motor. The paper would be considerably strengthened by focusing on the frequencies that are within the sensitivity range of this and other species and examining the effects of different sound amplitudes on the behavior.

Reviewer: 2

Comments to the Author(s)

This is a nice short paper in which the authors experimentally manipulated the acoustic environment and showed that the addition of white noise did not change the aggressive responses of cichlid groups towards a possible intruder, but did reduce affiliation and increased submissive behaviors of females within the groups. I found the paper easy to read and interesting, and the topic timely.

I have only one comment - the experimental design makes it impossible to tease apart the response of the fish to a possible competitor under noisy conditions from the response to the noisy conditions per se. The noise playback always started just before the fish were exposed to the competitors, and theoretically the reduced affiliation and increased submissive behavior could be a reaction of the fish to the new noisy conditions, regardless of the intruding fish. This can be at least partly dealt with by looking at the behavior of the group members in the minute that passed between the starting of the playback and the exposure the fish to the intruder.

===PREPARING YOUR MANUSCRIPT===

===PREPARING YOUR REVISION IN SCHOLARONE===

Author's Response to Decision Letter for (RSOS-202191.R0)

See Appendix A.

Decision letter (RSOS-210982.R0)

Dear Professor Radford

On behalf of the Editors, we are pleased to inform you that your Manuscript RSOS-210982 "Impacts of additional noise on the social interactions of a cooperatively breeding fish" has been accepted for publication in Royal Society Open Science subject to minor revision in accordance with the referees' reports. Please find the referees' comments along with any feedback from the Editors below my signature.

Please submit your revised manuscript and required files (see below) no later than 7 days from today's (ie 23-Jun-2021) date. Note: the ScholarOne system will 'lock' if submission of the revision is attempted 7 or more days after the deadline. If you do not think you will be able to meet this deadline please contact the editorial office immediately.

on behalf of Dr Isaac Ligocki (Associate Editor) and Kevin Padian (Subject Editor)
openscience@royalsociety.org

Associate Editor Comments to Author (Dr Isaac Ligocki):
Associate Editor
Comments to the Author:

Thank you for your efforts in making revisions and additions to your manuscript. These changes have addressed many of the reviewers comments and concerns.

My only minor comment on the revised manuscript is to consider more clearly stating our current understanding of the relationship between *N. pulcher* and *N. brichardi* (that they are different species). This distinction is not at all central to your study or its conclusions so I don't think more than a brief addition is necessary, but I could see a benefit to rephrasing your statement on Line 53 to indicate that these are considered different species. As you certainly know the relationships in this clade are complex, and as you state these names have been used interchangeably in the past. Nonetheless, clarifying that our current understanding of this group indicates they are different species may help avoid any confusion for readers unfamiliar with the system. You already cite Duftner et al. 2007 which makes this distinction; Gante et al. 2016, *Mol. Ecol.* is another more recent reference you may find helpful.

===PREPARING YOUR MANUSCRIPT===

===PREPARING YOUR REVISION IN SCHOLARONE===

Author's Response to Decision Letter for (RSOS-210982.R0)

See Appendix B.

Decision letter (RSOS-210982.R1)

Dear Professor Radford,

I am pleased to inform you that your manuscript entitled "Impacts of additional noise on the social interactions of a cooperatively breeding fish" is now accepted for publication in Royal Society Open Science.

on behalf of Dr Isaac Ligocki (Associate Editor) and Kevin Padian (Subject Editor)
openscience@royalsociety.org

Appendix A

Impacts of additional noise on the social interactions of a cooperatively breeding fish

Ines Braga Goncalves, Emily Richmond, Harry R. Harding & Andrew N. Radford

Response to Reviewers

We are glad that our substantial changes resolved the majority of the original issues raised by the reviewers, and are grateful for the opportunity to respond to the remaining few comments. We detail our responses in bold beneath each comment, with line numbers referring to the revised MS.

Reviewer: 1

I do not think the request to sound map more of the tank was unreasonable and could easily be accomplished. If the fish never moved from the spot where the sound was recorded, than that would be fine however the authors will need to show a resident time map of the fish location. The high level of variability shown in the study could be partially explained by large/sharp sound gradients in close proximity to the location where the sound was recorded.

The fish did move around their compartment during trials, so we agree that some consideration of sound variation in the tank is valid to our study. Within-group interactions can occur anywhere in the focal-group compartment; two of these behaviours (affiliation and submission) were shown to be affected by additional noise, so we do not believe that within-compartment variation in sound levels would explain the lack of an effect on the third behaviour (aggression). By contrast, defensive acts occur near to the intruder compartment; it is therefore possible, in principle, that the lack of an effect of additional noise on this behaviour is due to a greatly reduced sound level in this part of the tank. We have therefore conducted additional sound measurements as requested by the reviewer (lines 149–154). Whilst sound levels near the intruder compartment (which is furthest from the loudspeaker) are indeed lower than in the centre of the focal-group compartment and near to the loudspeaker compartment, they are still considerably higher than in the control treatment and likely audible to the study species (lines 154–157). This within-tank variation in sound level is thus unlikely to explain the lack of an effect of additional noise on defensive behaviour (lines 247–251).

The use of broad band sound from 0.1 to 20 Khz still remains unjustified. While I may agree that some sound caused some behavioral differences, I think it would be reasonable to understand what frequencies may have influenced behavior.

We agree that it would ultimately be interesting to know exactly what frequencies influence within-group behaviours, but that would require additional experiments. We have therefore added this as a point for future work in the Discussion (lines 278–281).

While I agree that ABRs have limited value, they are certainly useful to establish what frequencies that the fish could detect. The paper that reports this fish produces sound up to 12Khz plots the sound spectrum from 4 to 12kHz. And while one would think that sounds produce by animals are effective for intraspecific communication, this does not necessarily mean that the entire spectrum is detected. Additionally, these sounds are exceedingly short (10 ms) and their functional significance remains to be determined. Without compelling physiological or behavioral evidence to clearly indicate that species can detect frequencies above 5 Khz, most of the sound presentation in the frequencies < 5 Khz was limited to 100 dB which is probably why the behavioral differences were not clear.

It has been established using ABR that the study species, *Neolamprologus pulcher* (also referred to sometimes in the literature as *N. brichardi*; line 53), can detect sounds between 100 and 2000 Hz. But they are also reported capable of producing high-frequency sounds (two-pulsed calls with an average peak frequency of 12 kHz \pm 8 kHz) for communication (Spinks et al. 2017; our reference 31); they are therefore likely to have a hearing capacity to match. [With respect, this cited paper does not plot the sound spectrum from 4 to 12 kHz as

stated by the reviewer, but from 0 to 24 kHz (their Fig. 2B). Moreover, the authors report that, “When this double-pulse call occurred, often the first pulse had a dominant frequency between 7000 Hz and 15000 Hz and the second pulse peaked higher, above 17,000 Hz.” Given the more definite hearing capability of the study species up to 2 kHz, we have made sure that the emphasis is on this when we present the background information (lines 61–64) and have focused the majority of our presented acoustic measurements in that frequency range (lines 139–146, 154–157, Figure 1b). In the centre of the tank, root-mean squared sound-pressure level <2 kHz in the additional-noise treatment was 131.4–132.8 dB re 1 μ Pa. For any given frequency band within that, the power spectral density may be ca. 100 dB (Figure 1b), but the overall sound level is much greater. For illustrative purposes, we present in the main text just one set of values in the higher frequency range (4–20 kHz) that might be detected by the study species (lines 147–149). [NB Sound levels are now presented for 20-s recordings (lines 144–145) as that is the period of a single burst of additional noise; previously levels were presented for 30-s recordings, including the 5-s silence either side, but we have been advised by an acoustician that the former is more relevant.]

Also, the sound is claimed to be representative of transient motorboat but there is no evidence that this white noise is consistent with the spectrum from a boat motor.

We apologise for the confusion our writing created here. We were not seeking to claim that the white-noise playbacks were consistent with the spectrum from a boat motor; rather, that the temporal fade-in / fade-out element was equivalent to the coming and going (i.e. passing) of a motorboat. We have rephrased this sentence for clarity to be: “Each white-noise period had a 10 s fade-in and 10 s fade-out effect applied to reflect the temporal fluctuation in amplitude of, for example, a passing motorboat.” (lines 121–122).

The paper would be considerably strengthened by focusing on the frequencies that are within the sensitivity range of this and other species and examining the effects of different sound amplitudes on the behavior.

We have focused the majority of our sound measurements on the frequencies known for sure to be detected by the study species, but believe it is valid to provide for illustrative purposes the wider frequency range too (see detailed response to earlier comments). The investigation of different sound amplitudes on behaviour would require new experiments; we have therefore suggested this as future work in the Discussion (lines 278–281).

Reviewer: 2

I have only one comment – the experimental design makes it impossible to tease apart the response of the fish to a possible competitor under noisy conditions from the response to the noisy conditions per se. The noise playback always started just before the fish were exposed to the competitors, and theoretically the reduced affiliation and increased submissive behavior could be a reaction of the fish to the new noisy conditions, regardless of the intruding fish. This can be at least partly dealt with by looking at the behavior of the group members in the minute that passed between the starting of the playback and the exposure the fish to the intruder.

This is a good point. Unfortunately, we can’t analyse the behaviour in the one minute following initiation of playback and exposure of resident fish to the intruder because the video recording was started at the latter point (we have clarified this in the text; lines 175–176). However, the main result stands – noise affects within-group behavioural interactions in this outgroup context. We have added to the Discussion that we cannot tease apart the exact reason/mechanism for this; that would require further experiments (lines 278–281). We do not believe that detracts from the main conclusions, as the implications of noise-induced changes in within-group behaviour remain whatever the underlying mechanism for them.

Appendix B

Impacts of additional noise on the social interactions of a cooperatively breeding fish

Ines Braga Goncalves, Emily Richmond, Harry R. Harding & Andrew N. Radford

Response to Reviewers

We are glad that our paper has now been accepted for publication pending minor review. Below we explain our response to the one remaining query

Associate Editor

Comments to the Author:

Thank you for your efforts in making revisions and additions to your manuscript. These changes have addressed many of the reviewers comments and concerns.

My only minor comment on the revised manuscript is to consider more clearly stating our current understanding of the relationship between *N. pulcher* and *N. brichardi* (that they are different species). This distinction is not at all central to your study or its conclusions so I don't think more than a brief addition is necessary, but I could see a benefit to rephrasing your statement on Line 53 to indicate that these are considered different species. As you certainly know the relationships in this clade are complex, and as you state these names have been used interchangeably in the past. Nonetheless, clarifying that our current understanding of this group indicates they are different species may help avoid any confusion for readers unfamiliar with the system. You already cite Duftner et al. 2007 which makes this distinction; Gante et al. 2016, Mol. Ecol. is another more recent reference you may find helpful.

We have included clarification as suggested, and added the Gante et al. 2016 reference (lines 55–57).